# Integrative Analysis of Intrahepatic Cholangiocarcinoma Subtypes for Improved Patient Stratification: Clinical, Pathological, and Radiological Considerations

**DOI:** 10.3390/cancers14133156

**Published:** 2022-06-28

**Authors:** Tiemo S. Gerber, Lukas Müller, Fabian Bartsch, Lisa-Katharina Gröger, Mario Schindeldecker, Dirk A. Ridder, Benjamin Goeppert, Markus Möhler, Christoph Dueber, Hauke Lang, Wilfried Roth, Roman Kloeckner, Beate K. Straub

**Affiliations:** 1Institute of Pathology, University Medical Center of the Johannes Gutenberg-University Mainz, 55131 Mainz, Germany; mario.schindeldecker@unimedizin-mainz.de (M.S.); dirk.ridder@unimedizin-mainz.de (D.A.R.); wilfried.roth@unimedizin-mainz.de (W.R.); 2Department of Diagnostic and Interventional Radiology, University Medical Center of the Johannes Gutenberg-University Mainz, Langenbeckst. 1, 55131 Mainz, Germany; lukas.mueller@unimedizin-mainz.de (L.M.); christoph.dueber@unimedizin-mainz.de (C.D.); roman.kloeckner@unimedizin-mainz.de (R.K.); 3Department of General, Visceral and Transplant Surgery, University Medical Center of the Johannes Gutenberg-University Mainz, 55131 Mainz, Germany; fabian.bartsch@unimedizin-mainz.de (F.B.); lisa-katharina.groeger@unimedizin-mainz.de (L.-K.G.); hauke.lang@unimedizin-mainz.de (H.L.); 4Tissue Biobank, University Medical Center of the Johannes Gutenberg-University Mainz, 55131 Mainz, Germany; 5Institute of Pathology and Neuropathology, Hospital Ludwigsburg, 71640 Ludwigsburg, Germany; benjamin.goeppert@med.uni-heidelberg.de; 61st Clinic and Polyclinic of Internal Medicine, University Medical Center of the Johannes Gutenberg-University Mainz, Langenbeckst. 1, 55131 Mainz, Germany; markus.moehler@unimedizin-mainz.de

**Keywords:** liver cancer, subtyping, small duct type, large duct type, diagnostic imaging, adenocarcinoma

## Abstract

**Simple Summary:**

Liver cancer subtypes differ in prognosis and genetic alterations. An accurate diagnosis made on time is the key aspect of clinical decision-making. Hence, a correct diagnosis is of pivotal importance for individual patients. In this study, we identified the most relevant clinical, radiological, and histological parameters for an improved subtype diagnosis of intrahepatic cholangiocarcinoma. As a result of our study, the radiologist should consider factors such as growth pattern, location, and contrast agent behavior. For the pathologist, precursor lesions, mucin secretion, and a periductal-infiltrating growth are of utmost importance, while immunohistochemical analyses are essential for exclusion of extrahepatic malignancies, but have so far only value for iCCA subtype analysis in the context with other parameters.

**Abstract:**

Intrahepatic cholangiocarcinomas (iCCAs) may be subdivided into large and small duct types that differ in etiology, molecular alterations, therapy, and prognosis. Therefore, the optimal iCCA subtyping is crucial for the best possible patient outcome. In our study, we analyzed 148 small and 84 large duct iCCAs regarding their clinical, radiological, histological, and immunohistochemical features. Only 8% of small duct iCCAs, but 27% of large duct iCCAs, presented with initial jaundice. Ductal tumor growth pattern and biliary obstruction were significant radiological findings in 33% and 48% of large duct iCCAs, respectively. Biliary epithelial neoplasia and intraductal papillary neoplasms of the bile duct were detected exclusively in large duct type iCCAs. Other distinctive histological features were mucin formation and periductal-infiltrating growth pattern. Immunohistochemical staining against CK20, CA19-9, EMA, CD56, N-cadherin, and CRP could help distinguish between the subtypes. To summarize, correct subtyping of iCCA requires an interplay of several factors. While the diagnosis of a precursor lesion, evidence of mucin, or a periductal-infiltrating growth pattern indicates the diagnosis of a large duct type, in their absence, several other criteria of diagnosis need to be combined.

## 1. Introduction

Intrahepatic cholangiocarcinoma (iCCA) constitutes the second most common primary malignancy of the liver, with a poor median survival of only 13 months. Globally, many countries have experienced substantial increases in mortality in the past decades [1]. Resection is the only curative option, but only one-third of patients are diagnosed at a stage that still allows for surgical therapy [2]. Even if treated surgically, iCCAs have an unfavorable survival rate of 14% and 33%, respectively [3]. Locally advanced or metastatic iCCAs are treated with a combination of chemotherapeutic agents. However, therapies with chemotherapeutic agents show only minor survival benefits [4]. Currently, several personalized medicine approaches have been pursued in a second-line setting, yet are highly dependent on the identification of a molecular target [5]. iCCAs may be subdivided into two entities with distinct etiology, histology, molecular pathology, and separate cell type of origin. On the one hand, iCCAs may arise from progenitor cells of the canals of Hering, giving rise to small duct iCCAs, and on the other hand, from stem cells of peribiliary glands, so-called large duct iCCAs [6]. When compared, small duct iCCAs shows a better prognosis [7], possibly due to therapeutically relevant alterations such as *IDH1-/2*-mutations and *FGFR2* gene fusions [8,9]. Overall, both types reflect the molecular heterogeneity of iCCAs [10]. It has already been suggested that the lack of efficient therapeutic strategies may be the result of a lack of stratification according to the newly defined subgroups [5]. As aptly pointed out by Komuta, it is of crucial importance to correctly diagnose and subtype primary liver tumors in order to make the right therapeutic decisions [11].

This study aimed to unravel the best possible strategy to optimally subtype iCCAs as early as possible. The staging includes imaging techniques, which are often carried out before a biopsy is taken. For this purpose, we have characterized a large cohort of iCCAs. We comparatively analyzed iCCAs concerning their clinical parameters, radiological findings, and histomorphological and immunohistochemical results. For the best possible patient outcome, we will elucidate the most crucial factors which need to be addressed in imaging and histology. This manuscript may thus improve iCCA subtyping in clinical practice and help standardize the radiological and histopathological findings.

## 2. Results

### 2.1. Clinical Features of iCCA Subtypes

In order to unravel subtype-specific features of iCCAs, we established a large and clinicopathologically well-characterized cohort of 232 operation specimens of iCCAs, with 148 small duct iCCAs and 84 large duct iCCAs, diagnosed according to the current WHO classification (see Table 1).

Both subtype groups were comparable in terms of age and gender distribution as well as body mass index (BMI) and physical status (American Society of Anesthesiologists Physical Status Classification System, ASA) [12]. Large duct iCCAs were more often resectable, but often required major resections including the hilar bifurcation, and exhibited earlier and more frequent recurrences. These findings were consistent with a more central location and earlier clinical diagnosis. In addition, in large duct iCCAs, the initial symptom of jaundice was observed more frequently. However, only three cases of primary sclerosing cholangitis (PSC) (all large duct types) were included in our cohort.

### 2.2. Radiological Findings

Concerning radiology, cross-sectional imaging procedures from 191 of the 232 patients were available in sufficient quality, including 146 computed tomography (CT) and 45 magnetic resonance imaging (MRI) scans. While small duct iCCAs were on average slightly larger and less frequently located near the portal bifurcation, the distance to the serosa did not show a significant difference in the comparative analysis (Table 2). Large duct iCCAs showed a duct-accentuated growth pattern in 23 cases, whereas this occurred in only three cases of small duct iCCAs. The remaining cases were classified as mass-forming (Figure 1). This difference was significant in both univariate and multivariate analyses. Ductal growth pattern correlated significantly with biliary obstruction (r_s_ = 0.468, *p* < 0.001), hypovascular (r_s_ = 0.229, *p* = 0.002) and rim-enhanced (r_s_ = 0.227, *p* = 0.002) arterial enhancement pattern. For this reason, we excluded the parameters bile duct obstruction and contrast agent behavior from the regression analysis to avoid misinterpretation by multicollinearity. Other observed variables could not distinguish between both iCCA subtypes (Table 2). As expected, radiologic multifocality and pT2-stage (r_s_ = 0.319, *p* < 0.001), invasion of visceral peritoneum and pT3-stage (r_s_ = 0.596, *p* < 0.001), as well as invasion of extrahepatic structures and pT4 stage (r_s_ = 0.361, *p* < 0.001) were associated, respectively.

### 2.3. Histological and Immunohistochemical Features of iCCA Subtypes

Small and large duct iCCAs differed significantly in several histopathological characteristics (see Table 3 and Figure 2). Biliary intraepithelial Neoplasia (BilIn) and intraductal papillary neoplasms of the bile ducts (IPNB) were only diagnosed in large duct iCCAs. We observed mucin formation in 82% of large and only 12% of small duct iCCAs. A periductal-infiltrating (PI) growth was mainly found in large duct iCCAs (57% compared to 3%). However, due to the correlation between the presence of a precursor lesion and a PI growth pattern (r_S_ = 0.706, *p* < 0.001), the PI growth pattern was not an independent variable. Consequently, the growth pattern was not included in the regression analysis. Nevertheless, univariate analysis of growth patterns showed the third-highest effect size after the detection of precursor lesions and the production of mucin. A graphical representation of the difference in growth patterns is illustrated in Appendix B, Figure A1. Histological evidence of a desmoplastic stromal reaction, lymphangioinvasion and corresponding metastases, as well as perineural invasion showed considerably lower effect sizes. Those factors were more likely present in large duct iCCAs but sometimes diagnosed in small duct iCCAs, too.

In immunohistochemical analysis, the antibodies CA19-9, EMA, CD56, N-cadherin, and CRP showed potential in the differential diagnosis of iCCA subtypes, while CK7 and S100 did not (Table 3). CK20 was generally positive in large duct iCCAs. A positive staining reaction for CD56 was more likely indicative of small duct iCCAs. The significance of a positive staining reaction for N-cadherin and a negative EMA reaction was a comparable sign of small duct iCCAs. CA19-9 was rather unspecific and displayed a weak effect size. Each of the immunohistochemical stains showed a spectrum of reactions in both subtypes, and no single antibody alone was able to differentiate between both iCCA subtypes.

### 2.4. Patient Survival

Survival probability of the small duct iCCAs was better than that of the large duct iCCAs. However, in contrast to previous findings [13], this result was not statistically significant. This may be a secondary effect due to the low number of patients, who were eligible for the survival analysis, especially regarding the period after 5-year survival, or a bias due to the fact, that in our study, only resectable iCCAs were included. The median overall survival was 24.37 months for large duct iCCAs, compared to 35.17 months for small duct iCCAs (Figure 3).

## 3. Discussion

In our comprehensive study including 232 patients with iCCA, we unraveled the most important clinical, radiological, histological, and immunohistochemical features for the differential diagnosis of small and large duct iCCAs.

Clinical indicators of iCCA are limited, and rarely reliable indicators to establish the iCCA subtype diagnosis. Initial presentation with jaundice favors the diagnosis of a centrally located tumor, yet may not distinguish iCCA from perihilar cholangiocarcinoma (pCCA) or benign diseases. When compared to small duct iCCAs, large duct iCCAs more often required larger and more invasive surgery involving the hilar region.

Imaging techniques, especially CT and MRI, naturally play a pivotal role in the management of iCCAs. Various radiologic approaches have been proposed to distinguish iCCA subtypes. However, subtype comparisons are complicated by the fact that some even recently published studies only distinguish between mass-forming iCCAs and non-mass-forming iCCAs without considering the final histopathological subtype [14,15,16,17]. In general, large duct iCCAs more often shows a ductal accentuated growth pattern with consecutive biliary obstruction [18]. On CT and MRI, large duct iCCAs typically presents as a periductal thickening and increased enhancement. In this study, we were able to provide evidence that a periductal accentuated growth pattern and the presence of biliary obstruction are the radiologically decisive factors for iCCA subtype classification. In this regard, periductal growth has been demonstrated to be strongly associated with biliary obstruction. These findings are consistent with the literature, but the present study is, to our knowledge, the first to analyze the exact histological subtypes in a blinded fashion. Akita et al. showed that the distance of tumor to portal bifurcation was largest in small duct iCCAs when compared to large duct iCCAs or pCCA [19]. In the present study, we were able to validate this observation in a larger cohort, although this measure can only serve as an approximate indicator. The tumor growth pattern was more important. Mass-forming-type iCCAs are mostly hypovascular or rim-enhanced [20,21,22]. Fujita et al. demonstrated that, in the arterial phase, tumors that showed the characteristics of large duct iCCAs had a hypovascular enhancement pattern [23]. However, in contrast to previous studies, we were not able to demonstrate a dominating arterial enhancement pattern. Most small- and large duct iCCAs present a hypovascular enhancement. Thus, there is no sufficient selectivity to differentiate iCCA subtypes. However, we detected a correlation between the periductal growth pattern and hypovascular as well as rim-enhanced pattern. This may indicate that the arterial enhancement pattern is more dependent on the individual growth pattern than the actual histological subtype.

Taking into account the frequency of liver lesions, studies involving comprehensive clinicopathologic investigations on the differential diagnoses are of special importance. Apart from the frequent benign liver hemangioma, rare cases of hepatocellular adenoma and focal nodular hyperplasia, primary and secondary malignancies need to be ruled out. In addition to iCCAs, also hepatocellular carcinomas (HCCs) are primary liver carcinomas with characteristic clinical and radiologic features distinguishing both entities. The hepatocellular origin of a tumor may already be distinguished conventionally and morphologically by routine histopathology together with typical radiologic features and clinical context, as HCCs are most frequently associated with liver cirrhosis. In particularly difficult cases, as for certain types of HCCs as well as combined hepatocellular-cholangiocarcinoma (HCC-CCC), the diagnosis needs further immunohistochemical analyses such as Hepatocyte paraffin 1 (HepPar1) [24]. So far, the final histopathologic diagnosis of iCCA including its subtype is necessary for further therapy planning, even in unresectable cases [25]. Tumor biologic heterogeneity is reflected in the new classification of subtypes, which incorporates a large number of different diagnostic factors. The exact determination of iCCA subtypes is not only the basis for further studies but also for the individual patient prognosis and therapeutic regimens. For these reasons, one of the central goals of the European Cholangiocarcinoma Network is to determine factors important for subtyping [26]. Numerous features have already been identified that distinguish iCCA subtypes on a histologic basis. These characteristic histologic features include precursor lesions [27,28], growth patterns [29], perineural invasion and lymph node metastases [30], stromal desmoplasia, and mucin [8]. In our analyses, we demonstrated that some factors outclass others in iCCA subtype differentiation. Precursor lesions were found exclusively in large duct iCCAs and mucin production was negative in 88% of small duct iCCAs compared with the 18% in large duct iCCAs. In addition, we were able to show that not only the macroscopic growth pattern—which may not even be known—but also the microscopic growth pattern may significantly help iCCA subtyping. The strong correlation between a periductal-infiltrating growth pattern and the detection of precursor lesions suggests a direct relationship. It may be assumed that bile ducts with BilIns high grade act as origin of the invasive carcinoma outgrowth. In some cases, however, no precursor lesion may be seen, as the bile duct, from which the invasive carcinoma originates, has been completely consumed. As large bile ducts and peribiliary glands often produce mucin, also large duct iCCAs may produce mucin in contrast to small duct iCCAs [31]. An iCCA that does not show a precursor lesion, mucin production, or a periductal-infiltrating pattern can only be designated as a large duct iCCA by a combination of several other histological criteria. Factors such as stromal desmoplasia, lymphangioinvasion, and lymph node metastases, especially in the case of small tumors, perineural infiltration, and a central location should then be considered. In these cases, it would also be advisable to determine the diagnosis conclusively only in the context of an interdisciplinary tumor board. In rare cases, the classification into the iCCA subtypes was difficult as some borderline cases showed properties of both entities, possibly representing a kind of intermediate duct iCCA. For the sake of this study, we assigned the specific subtype for which they presented the most characteristic features, although not all characteristics were fulfilled.

Immunohistochemistry should be used in unclear cases, especially beforehand, when a metastatic event must be differentiated from a primary liver tumor. However, immunohistochemical analysis may be used in iCCA subtype analysis. This has been shown several times for several different markers. In a recent study involving a small cohort, CRP and N-cadherin-positivity as well as S100-negativity supported the diagnosis of small duct iCCAs [32]. We were able to validate this in our much larger cohort for CRP and N-cadherin, whereas, in our hands, the presence/absence of S100 was not able to differentiate between these subtypes. It is worth mentioning that we did not use the same S100 antibody as in the original publication, which may explain the discrepancy. In addition, positive immunohistochemical staining of EMA, CA19-9, and CK20 was indicative of large duct iCCAs. CD56 is most often positive in small duct iCCAs. EMA-positivity in iCCA subtypes has so far only been shown for the mucinous variant [33], which belongs to large duct iCCAs according to the current WHO classification. To our knowledge, an analysis of CK20 in the differentiation of iCCA subtypes has not yet been reported in the literature. CA19-9 is a widespread serum biomarker, but it may be used immunohistochemically to facilitate differential diagnosis. CA19-9 is most often expressed in ductal adenocarcinomas of the pancreas, but may be positive in many other entities as well [34]. In our series, CA19-9 was preferably positive in large duct iCCAs, which shares similarity to ductal adenocarcinoma of the pancreas. CD56 has been demonstrated to be characteristic for a cholangiocellular component of combined hepatocellular and cholangiocarcinoma, and thus, in small duct iCCAs [35].

The present study has several limitations. First, the retrospective data acquisition of a pure operation cohort may have inadvertently led to a selection bias. However, this cannot be entirely avoided due to the rarity of iCCAs, so we collected a fairly large patient cohort to obtain the most comprehensive evaluation possible. Second, in contrast to the radiologist, the pathologist was not blinded by the subtype. Since the histological parameters were evaluated on the whole slides of the respective cases, the individual data may have been subconsciously interpreted. Nevertheless, since the immunohistochemical analyses were carried out at the TMA, blinding was guaranteed. Third, we compared MRI and CT images, which may have affected the results. CT is considered the standard imaging method for the assessment of liver masses, but is only of limited use for determining the exact tumor extent along bile ducts, and hence, the resectability [36]. Due to the high spatial resolution of CT and the better representation of soft tissues, in iCCAs, MRI is more sensitive but less specific than CT [15]. Overall, however, our approach primarily reflects clinical reality. Fourth, we did not address the differential diagnosis of hepatocellular carcinoma or metastatic adenocarcinoma in the frame of the present study. A detailed description of these and the possible implications would go beyond the scope of this paper. Fifth, due to the changes to the current classification of tumors, most previous research did not uniformly diagnose the subtypes. Although the findings of several studies presented were integrated into the categorization of the subtypes themselves, the exact entity may not have been thoroughly defined. However, this is also a strength of the present study, to characterize the subtypes for the first time using different modalities within a comprehensive study.

## 4. Conclusions

Small and large duct iCCAs are two biologically different tumor entities, and their correct classification is therefore of utmost importance for prognosis and subsequent therapeutic decisions.

In our analyses, growth pattern and presence of biliary obstruction were important for radiologic iCCA subtype diagnosis. For the final histopathologic diagnosis, a periductal-infiltrating growth pattern, mucin production, and precursor lesions such as BilIn and IPNB are the main factors that support the diagnosis of large duct iCCAs. If these factors are not present, the diagnosis of a large duct iCCA should only be made by combining various other factors, possibly including radiological and immunohistochemical analyses.

## 5. Materials and Methods

### 5.1. Patient Cohort

The present retrospective cohort study includes 232 patients diagnosed with iCCA between January 2006 and August 2021 at the University Medical Center Mainz, Institute of Pathology. The iCCA subtypes were diagnosed based on typical macroscopic growth and localization as well as combination of histopathological parameters. Detailed clinical course data are continuously maintained in an existing relational Access database (Microsoft Corporation, Redmond, WA, USA). For the current analysis, we retrieved baseline characteristics, survival data, concomitant diseases, and surgical characteristics. We assessed complication rates according to the Dindo–Clavien classification, with major complications defined to be requiring at least surgical, endoscopic, or radiologic intervention [37].

### 5.2. Radiologic Workup

Preoperative imaging data were available from 191 out of 232 patients. Contrast-enhanced CT and MRI were performed in 146, and 45 cases, respectively. CT studies were routinely performed using a 256-, 64-, and 16-slice CT scanner (iCT, Brilliance 64, Brilliance 16, Philips, Eindhoven, Netherlands). Iomeprol (Imeron 400, Bracco, Milano, Italy) was used as a contrast medium with a body weight-adapted dosage (1.5 mL/kg body weight) and injection rate of 4 mL/s using an automated injector (Accutron CT-D^®^, Medtron, Germany). The patients underwent an abdominal CT scan with an arterial phase tube voltage of 80 kV or 100 kV and a venous phase tube voltage of 120 kV. Tube current was modulated between 120 and 800 mA. Axial images were acquired with a slice thickness of 0.625 mm. During post-processing, images with a slice thickness of 1 mm, 3 mm, and 5 mm were reconstructed in the axial orientation and in sagittal and coronal views. The MRI was performed with different scanners: 1.5 T Sonata^®^, 1.5 T Avanto^®^, 3 T Trio^®^, 3 T Skyra^®^ or 3 T Prisma ^®^ (all Siemens Healthcare, Erlangen, Germany). All patients underwent a similar imaging protocol comprising of four dynamic, contrast-enhanced, T1-weighted fat-saturated, three-dimensional acquisitions and a delayed T1-weighted fat-saturated transversal acquisition. The contrast agents were Magnevist^®^ (gadolinium-diethylenetriaminepentacetate; Bayer Schering Pharma AG, Germany), Dotarem^®^ (gadolinium-1,4,7,10-tetraazacyclododecane-1,4,7,10-tetraacetic acid; Guerbet, France) or Primovist^®^ (gadoxetate disodium; Bayer Schering Pharma AG, Germany), which were applied with a body weight-adapted dosage (1.5 mL/kg body weight) and injection rate of 4 mL/s using an automated injector (Accutron MR^®^, Medtron, Germany). We accepted externally produced CT or MRI if it was of sufficient quality. The radiologists were blinded from the histological subtype of iCCA. In patients with available CT images, the average Hounsfield units (HU) of liver and tumor tissue were measured in a central region of interest. Tumor density was standardized relative to background liver density, both measured in HU during the late arterial phase. We classified the growth pattern as either mass-forming or ductal-growing. The ductal-growing pattern was defined by a growth pattern at or around existing bile ducts. The relationship between the portal vein and hepatic artery was also assessed and categorized as follows: no contact, invasion, and entrapment. The minimal tumor-free distance between tumor and serosa as well as the portal bifurcation was measured.

### 5.3. Histology and Immunohistology

All formalin-fixed paraffin-embedded (FFPE) whole-slide tissue samples were histologically reevaluated for the presence of given histological parameters and classified into small- and large duct iCCAs according to the WHO classification of tumors of the digestive system (fifth edition, 2019) [7]. The diagnosis, as well as the subtype, were determined after viewing all section specimens by two experienced pathologists with special expertise in the field of liver pathology (B.K.S. and T.S.G). Every criterion mentioned by the WHO classification to distinguish both subtypes was equally observed. Only cases classified as iCCA both histologically and during the course were included. Other entities were excluded. Doubtful iCCA cases were jointly investigated and discussed until consensus was reached.

Precursor lesions were classified accordingly. Depending on tumor location, as previously stated, we subdivided perihilar cholangiocarcinoma (pCCA) from iCCA [38]. pCCA was defined as primarily located in the hilus region at the common hepatic duct, possibly extending to the right or left hepatic duct. All tumors were restaged according to the Union for International Cancer Control (eighth edition with corrections from 2020) [39]. The tissues were examined with hematoxylin and eosin (HE), and Periodic acid–Schiff (PAS) staining. The presence of lymphatic invasion, vascular invasion, and perineural invasion was defined by tumor cells in the lumen of vessels or adjacent to the nerve bundles, respectively. Tumor necrosis was defined as necrotic cells adjacent to viable tumor tissue and categorized into the four groups: no necrosis, little necrosis (<10%), moderate necrosis (10–50%), and extensive necrosis (>50%). We categorized the amount of cancer cell mucin generation into four groups: no mucin, little mucin (visible only at 400x magnification or in Pas-staining), moderate mucin (easily visible in 200x magnification), and extensive mucin (more than 50% of tissue area mucinous). The desmoplastic stromal reaction was classified into minimal (tumor cells without considerable intervening stroma), moderate (tumor area mostly comprised of tumor cells with some intervening stroma), and extensive (tumor area comprised of more intervening stroma than tumor cells). Based on the previously described gross classifications of Aishima et Oda [29], we categorized each case microscopically in mass-forming (solid tumor area), intraductal-growing (tumor mass in bile ducts), and periductal-infiltrating (bile ducts with irregular circularly tumor propagations). In compliance with the respective tumor heterogeneity, multiple entries were allowed. When all histological parameters, as well as tumor cell morphology, were taken into account, the iCCA subtype was determined. Due to the wide range of tumor cell morphology, a semiquantitative representation was omitted. Tissue microarrays (TMA) were generated as previously described [38]. The collection of human tumor samples and data were in compliance with the ethical guidelines of the Declaration of Helsinki and approved by the Institutional Ethics Committee (permit numbers 837.280.17 (11114) and 2021-15819).

Immunostaining was performed on TMA blocks sliced into 2–4 µm thick sections, deparaffinized, and rehydrated with decreasing concentrations of ethanol. Antigen retrieval and staining were conducted according to the manufacturer’s instructions (for more details, see Appendix A). Only primary iCCAs were used for the survival analysis of our cohort. Cases with irresectability, neoadjuvant treatment, and perioperative mortality were excluded (defined as death within the first 30 days [40]).

### 5.4. Statistical Analysis

Survival analyses were plotted by the Kaplan–Meier method and compared by the log-rank test. To compare the distribution of categorical variables, we conducted chi-square tests using SPSS 27 Software (SPSS, Chicago, IL, USA). In addition, Cramér’s V was calculated and interpreted as follows: 0.1–0.3: small, 0.3–0.5: moderate, and >0.5: large effect size [41]. We calculated the Spearman rank-order (r_S_) correlation coefficient for ordinal data. For multivariate analysis, the R environment for statistical computing (version 3.6.3, (R Foundation for Statistical Computing, Vienna, Austria)) and the Bias-Reduced Logistic Regression Statistics Software (maximum penalized likelihood, (Bias Reduced Logistic Regression (v1.0.6) in Free Statistics Software (v1.2.1), Office for Research Development and Education, (http://www.wessa.net/rwasp_logisticregression.wasp/)) (accessed on 2 May 2022) as proposed by Firth was used [42,43,44]. A separate regression analysis was conducted for clinical, radiological, pathological, and immunohistochemical data. We included all variables, regardless of significance level in the univariate analysis, unless otherwise noted. Overall survival was defined as the interval between initial diagnosis and death or last follow-up. A *p*-value of ≤0.05 was considered statistically significant. Venn diagram has been visualized using the Venny Software (VENNY. An interactive tool for comparing lists with Venn Diagrams, https://bioinfogp.cnb.csic.es/tools/venny/index.html. (accessed on 2 May 2022)) [45].

## Figures and Tables

**Figure 1 cancers-14-03156-f001:**
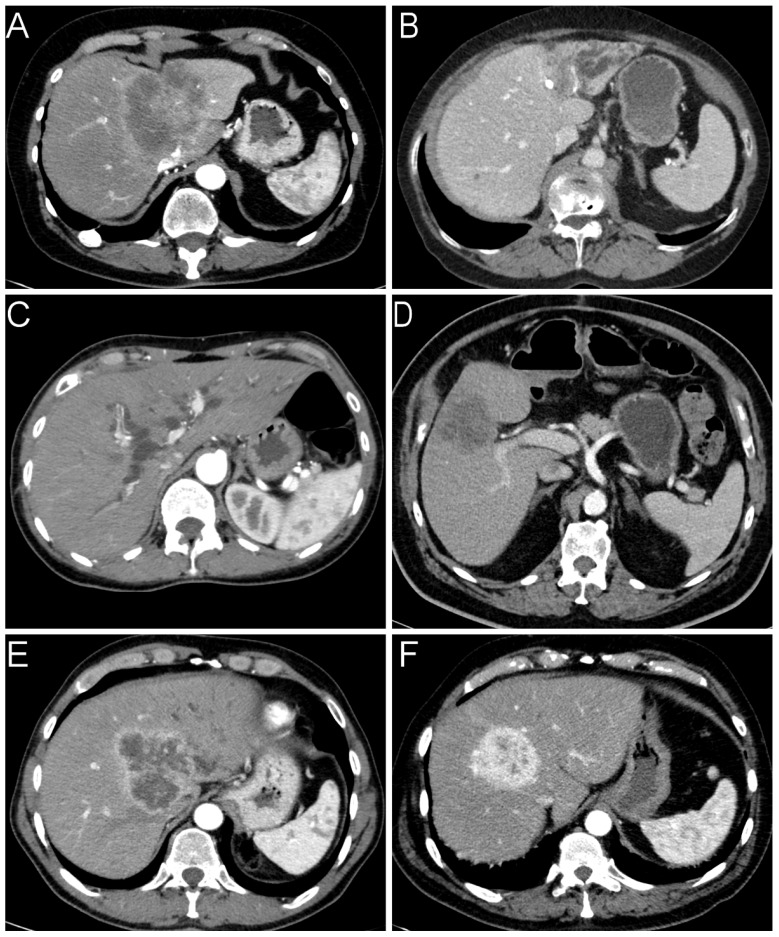
Radiological features of iCCAs. We differentiated two types of growth patterns: (**A**) mass-forming and (**B**) ductal-growing (in this case with an inserted biliary drainage). (**C**) Obstruction of bile ducts. Contrast agent behavior in the late arterial phase: (**D**) hypodense; (**E**) rim-enhanced; (**F**) hyperdense.

**Figure 2 cancers-14-03156-f002:**
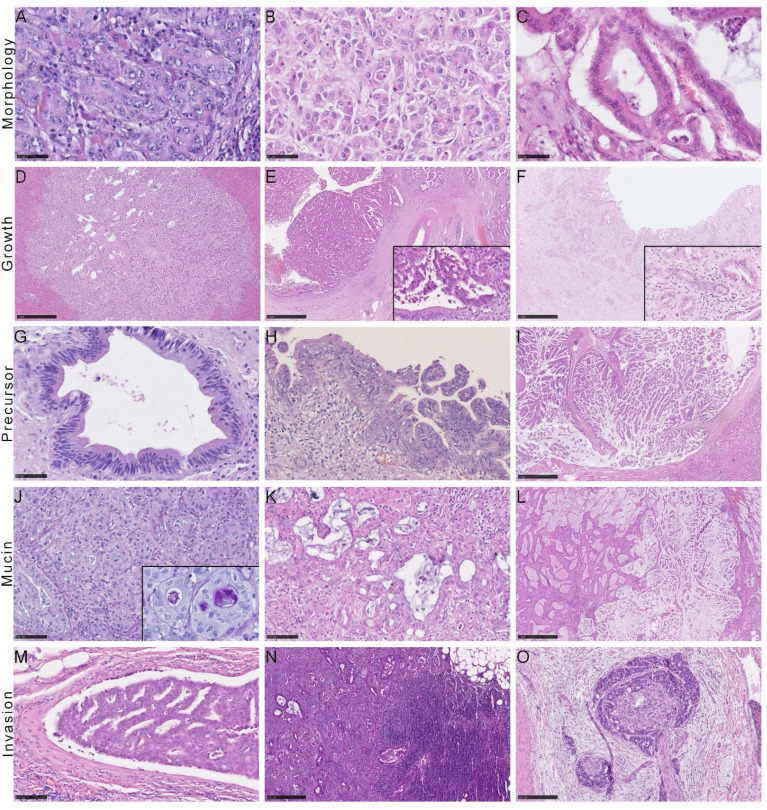
Histological distinction features of iCCA. (**A**) Adenocarcinoma forming solid sheets; (**B**) with cuboidal cells; (**C**) Infiltrative glands composed of prismatic cells. Different histological growth patterns of iCCA showing (**D**) a mass-forming (MF) tumor, (**E**) a tumor showing intraductal growth (IG), and (**F**) a tumor with periductal-infiltrating growth. Note the adjacent inconspicuous biliary epithelium (inlay; (**E**,**F**)). (**G**) Biliary intraepithelial neoplasia of low-grade and (**H**), 200X magnification) high-grade; (**I**) intraductal papillary neoplasm of the bile ducts. Different amounts of mucin: (**J**) little mucin with PAS-positivity showing in the inlaid picture; (**K**) moderate and (**L**) extensive amounts of mucin. (**M**) Lymphangioinvasion and (**N**) lymph node metastasis. (**O**) Perineural invasion.

**Figure 3 cancers-14-03156-f003:**
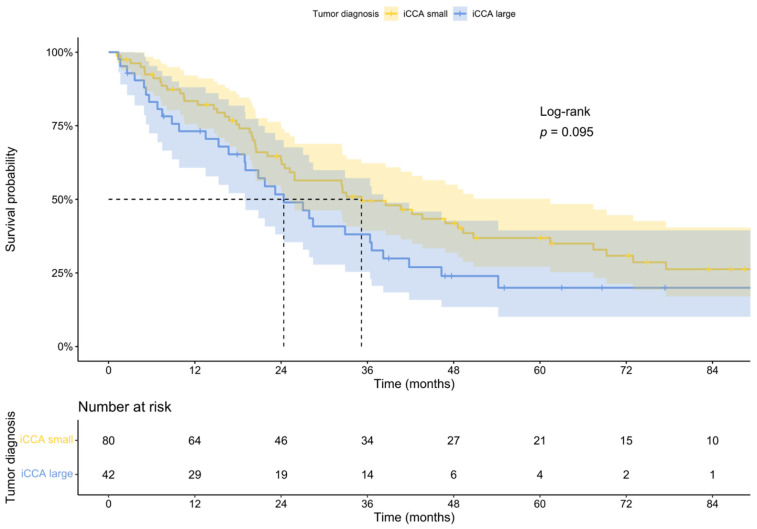
Overall survival of patients according to different iCCA subtypes.

**Table 1 cancers-14-03156-t001:** Clinical patient characteristics.

Patient Characteristics		Small Duct iCCAs *n* = 148	Large Duct iCCAs *n* = 84	Univariate		Multivariate		
				*p*-Value	Effect Size	*p*-Value	OR	95% CI
**Clinical**	Age [sd (range)]	65.64 (32–84)	62.37 (34–83)					
	Male/Female [%]	52/48	56/44					
	ASA I/II/III/IV [%]	1/46/51/2	1/43/52/4					
	BMI [mean ± sd]	26.86 ± 4.42	26.58 ± 5.03					
	Jaundice at diagnosis [%]	8	27	**0.005**	**0.260**	0.136	3.70	0.64–21.5
	Cardial/ pulmonal/	22/11/5	24/5/3					
	renal comorbidities [%]							
	Diabetes mellitus [%]	24	15	0.16		0.583	1.37	0.44–4.31
	Alcohol abuse [%]	4	7	0.573		0.749	0.67	0.06–8.02
	Nicotine abuse [%]	30	15	0.068		0.299	0.53	0.15–1.81
	PSC [%]	0	3	**0.021**	**0.152**	0.202	12.18	0.24–610.39
	Viral Hepatitis [%]	1	4	0.945		0.491	2.76	0.14–52.94
**Surgical**	Neoadjuvant treatment [%]	9	11					
	RFA/TACE/	8/8/0/84	0/12/0/88					
	SIRT/ Chemotherapy [%]							
	Initially irresectable [%]	19	7	**0.018**	**0.166**	0.325	4.59	0.21–101.85
	Major liver resection [%]	54	76	**0.002**	**0.222**	0.416	1.56	0.52–4.64
	Resection of hilar bifurcation [%]	10	33	**<0.001**	**0.279**	0.316	1.98	0.51–7.73
	Lymphadenectomy [%]	71	78	0.3032		0.416	0.63	0.20–1.96
	Major complication [%]	34	32	0.735		0.543	0.71	0.23–2.21
	Recurrence [%]	51	60					
	Months to recurrence [mean ± sd]	15.66 ± 23.51	11.24 ± 11.05					

sd: standard deviation. OR: odds ratio. CI: confidence interval. RFA: radiofrequency ablation; TACE: transarterial chemoembolization; SIRT: selective internal radiation therapy. Bold font indicates statistical significance.

**Table 2 cancers-14-03156-t002:** Correlation of iCCA subtypes and radiologic findings.

Tumor Characteristics		Small Duct iCCAs *n* = 122	Large Duct iCCAs *n* = 69	Univariate		Multivariate		
				*p*-Value	Effect Size	*p*-Value	OR	95% CI
**Radiology**	CT/ MRT [%]	79/21	72/28					
	Tumor size [mean ± sd]	7.79 ± 3.73	6.23 ± 2.30					
	Ductal growing pattern [%]	2	33	**<0.001**	**0.446**	**<0.001**	**16.17**	**3.07–85.06**
	Obstruction bile ducts [%]	25	48	**0.001**	**0.237**			
	Vascular invasion [%]	20	25	0.586		0.997	1.00	0.34–2.98
	Portal vein			0.229		0.939	1.02	0.56–1.87
	invasion [%]	17	18					
	entrapment [%]	20	29					
	Hepatic artery			0.713		0.156	0.62	0.32–1.21
	invasion [%]	3	1					
	entrapment [%]	16	12					
	Distant metastasis [%]	7	6	0.677		0.710	0.75	0.16–3.58
	Distance portal bifurcation [mean ± sd]	2.98 ± 2.37	1.56 ± 2.20					
	≥0.5 cm [%]	58	41	**0.039**	**0.151**	0.240	0.66	0.23–1.86
	Distance serosa [mean ± sd]	0.38 ± 0.82	0.43 ± 0.88					
	≥0.5 cm [%]	16	20	0.192		0.812	0.86	0.25–2.98
	Tumor density [mean ± sd]	0.97 ± 0.34	0.91 ± 0.39					
	Contrast agent behavior							
	hypovascular [%]	44	65	**0.006**	**0.204**			
	rim-enhanced [%]	28	18	0.154				
	hypervascular [%]	28	17	0.074				
	Multifocal [%]	31	28	0.600		0.622	1.25	0.51–3.07
	Invasion visceral peritoneum [%]	20	28	0.211		0.622	1.34	0.41–4.32
	Invasion extrahepatic structures [%]	8	13	.282		0.786	1.23	0.27–5.60
	Radiological diagnosis							
	HCC/iCCA/ primary liver tumor/	1/63/6/1/10	1/68/19/0/12					
	metastasis/not classified [%]							

Measurements are displayed in cm. sd: standard deviation. OR: odds ratio. CI: confidence interval. Bold font indicates statistical significance.

**Table 3 cancers-14-03156-t003:** Histology and immunohistochemistry of iCCA subtypes.

Tumor Characteristics		Small Duct iCCAs *n* = 148	Large Duct iCCAs *n* = 84	Univariate		Multivariate		
				*p*-Value	Effect Size	*p*-Value	OR	95% CI
**Pathology**	Tumor size [mean ± sd]	7.66 ± 4.03	6.31 ± 3.01					
	BilIn/IPNB [%]	0/0	64/ 11	**<0.001**	**0.731**	**<0.001**	**91.69**	**5.71–1472.03**
	BilIn low/high-grade [%]		25/ 75					
	Mass-forming [%]	99	78	**<0.001**	**0.342**			
	Intraductal-growing [%]	15	12	0.156				
	Periductal-infiltrating [%]	3	57	**<0.001**	**0.605**			
	Multifocal [%]	32	19	**0.028**	**0.144**	0.251	0.47	0.13–1.74
	Mucin [%]	88/10/2/0	18/56/24/2	**<0.001**	**0.696**	**<0.001**	**13.70**	**4.31–43.55**
	Necrosis [%]	36/27/29/8	25/32/37/6	0.131		0.599	1.22	0.57–2.58
	Desmoplasia [%]	36/55/9	14/67/19	**<0.001**	**0.282**	0.011	4.41	1.37–14.23
	Liver steatosis [mean ± sd]	10.00 ± 15.39	12.98 ± 21.14					
	Liver fibrosis [%]	49/34/11/1/4	47/35/13/1/4					
	UICC I/II/III/IV [%]	36/30/28/6	32/12/46/10					
	pT1/pT2/pT3/pT4 [%]	46/42/7/6	43/31/14/12					
	pNX [%]	18	20					
	pN0/pN1 [%]	58/24	39/41	**0.010**	**0.199**	0.152	2.62	0.68–10.03
	G1/G2/G3/G4 [%]	1/73/25/1	1/68/31/0					
	L1 [%]	13	28	**0.006**	**0.189**	0.092	3.78	0.78–18.35
	Lymph node metastases [mean (range)]	2.32 (1–7)	3.09 (1–16)					
	V1-2 [%]	22	23	0.068		0.877	1.12	0.27–4.59
	Pn1 [%]	16	40	**<0.001**	**0.293**	0.666	1.36	0.33–5.63
	R0/R1-2 [%]	81/15	73/26	**0.002**	**0.253**	0.786	1.23	0.27–5.52
	Margin distance [mean ± sd]	0.47 ± 0.60	0.48 ± 0.60					
**IHC**	CK7 [%]	1/8/91	1/4/95	0.369		0.149	2.94	0.66–13.10
	CK20 [%]	77/22/1	50/41/9	**<0.001**	**0.218**	**0.020**	**2.30**	**1.12–4.72**
	Ca19-9 [%]	12/53/35	4/46/50	**0.027**	**0.177**	0.186	1.54	0.80–2.95
	EMA [%]	4/39/56	0/25/75	**0.008**	**0.204**	0.055	2.13	0.97–4.69
	S100 [%]	72/26/2	76/24/0	0.498		0.792	1.23	0.25–5.96
	CD56 [%]	39/45/16	61/33/6	**0.003**	**0.226**	**0.008**	**0.45**	**0.25–0.82**
	N-cadherin [%]	13/31/56	27/33/30	**0.004**	**0.262**	0.124	0.67	0.40–1.13
	CRP [%]	7/16/77	24/26/50	**<0.001**	**0.290**	0.101	0.63	0.36–1.11

Measurements are displayed in cm. sd: standard deviation. OR: odds ratio. CI: confidence interval. Mucin/necrosis: none/little/moderate/extensive. Desmoplasia: minimal/moderate/extensive; Liver fibrosis: none/portal/septal/septal with distorted architecture/cirrhosis; Immunohistochemistry: minimal/weak/strong staining. Bold font indicates statistical significance.

## Data Availability

The data that support the findings of this study are available on request from the corresponding author.

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
