# Peer review of "Integrative Analysis of Intrahepatic Cholangiocarcinoma Subtypes for Improved Patient Stratification: Clinical, Pathological, and Radiological Considerations"

_cancers, 2022, doi:10.3390/cancers14133156_

Round 1

Reviewer 1 Report

Thank you for submitting this paper. The paper is well-written. The paper is technically sound, I have some comments to the authors. 

Line 30: The authors mention desmoplasia as a major factor, but later they mainly talk about precursor lesions, mucin, and the growth pattern. Could the authors align the key statements?

Line 126 and Line 136: It seems like spaces are missing.

Line 127-129: A periductal infiltrating growth pattern and precursor lesions correlate very strongly. Can the authors explain why this might be so?

Lines 160-166: Survival analysis: Could the authors comment on whether overall survival differs from previous study results, and if so, why?

Line 208: HCC: regarding the abbreviation, the plural form is inconsistent.

Line 208-212: In this section, the authors explain how the histological differential diagnosis of hepatocellular carcinomas is made. However, it is not clear exactly how this important differential diagnosis is excluded. Could this point be elaborated on?

Line 223-225: Mucin appears to be an important factor. Do the authors have an explanation for the fact that mucin is secreted less frequently in small duct iCCA?

Table 1: No statistical analysis was reported for cardiac, pulmonary, and renal comorbidities. However, appropriate evaluations were performed for alcohol abuse, nicotine abuse, and diabetes mellitus. Could the authors explain why?

Appendix Figure 1: From the figure, it can be seen that the histological growth can take several forms, in contrast to the radiological growth pattern. However, there is no comment on this in the methods section. Could the authors please explain why they chose this approach?

Author Response

Thank you for submitting this paper. The paper is well-written. The paper is technically sound, I have some comments to the authors.

Line 30: The authors mention desmoplasia as a major factor, but later they mainly talk about precursor lesions, mucin, and the growth pattern. Could the authors align the key statements?

We thank the reviewer for the valuable comments on behalf of our article! Stroma desmoplasia is indeed an important factor, but in our analyses, there are more decisive findings that may therefore better help differentiate between the two subtypes. For this reason, we have replaced desmoplasia in the simple statement with a periductal infiltrating growth pattern.

Line 126 and Line 136: It seems like spaces are missing.

We thank the reviewer for this observation, we have corrected spacing.

Line 127-129: A periductal infiltrating growth pattern and precursor lesions correlate very strongly. Can the authors explain why this might be so?

The reviewer has raised a crucial point. We included an additional explanatory sentence in the discussion. Shortly, large duct type iCCAs seem to preferentially develop from BilIn lesions and thereby are preferentially detected near the larger bile ducts they derive from. In this respect, we have also improved figure 2H by a novel image better presenting high grade BilIN (3) and refer to the subject in the graphical abstract.

Lines 160-166: Survival analysis: Could the authors comment on whether overall survival differs from previous study results, and if so, why?

The results of the survival analysis are concordant with the expected outcomes. Due to the composition of our cohort and the number of patients, there was no significance at the p<.05 level.

Line 208: HCC: regarding the abbreviation, the plural form is inconsistent.

We have corrected this accordingly.Line 208-212: In this section, the authors explain how the histological differential diagnosis of hepatocellular carcinomas is made. However, it is not clear exactly how this important differential diagnosis is excluded. Could this point be elaborated on?

We thank the reviewer for his suggestion and have now included  an additional sentence in this paragraph.

Line 223-225: Mucin appears to be an important factor. Do the authors have an explanation for the fact that mucin is secreted less frequently in small duct iCCA?

The reviewer is correct. Large duct iCCA display features of the cholangiocytes of the large bile ducts as well as peribiliary glands often also demonstrating mucus formation.

Table 1: No statistical analysis was reported for cardiac, pulmonary, and renal comorbidities. However, appropriate evaluations were performed for alcohol abuse, nicotine abuse, and diabetes mellitus. Could the authors explain why?

We expected that alcohol consumption, and presence of diabetes mellitus might have a potential influence on subtyping, as small duct iCCA may preferentially be more frequent in liver cirrhosis.

Appendix Figure 1: From the figure, it can be seen that the histological growth can take several forms, in contrast to the radiological growth pattern. However, there is no comment on this in the methods section. Could the authors please explain why they chose this approach?

In our experience, the growth pattern of iCCA, even within a single case, is often too diverse to be reduced to just the dominant growth pattern. For this reason, we have decided to document all existing growth patterns.

Reviewer 2 Report

Dear colleagues, thank you for your analysis of subtyping cholangiocarcinoma.

I only have two minor notes:

(1) in Chapter 2.3 Histological and immunohistochemical features...

"In immunohistochemical analysis, the antibodies CA19-9, EMA, S100, CD56, N-cadherin, and CRP showed potential in the differential diagnosis of iCCA subtypes, while CK7 and S100 did not (Table 3)."

S100 is listed in both arguments, whereas in Table 3, S100 Antibody is not shown as a helpful discriminator. Please clarify.

(2) in Chapter 3

"the diagnosis needs further l immunohistochemical analyses such as Hepatocyte paraffin 1 (HepPar1) [25]. "

I would expect the "l" being a typo

Congratulation to the nice piece of work,

Best regards

Author Response

Dear colleagues, thank you for your analysis of subtyping cholangiocarcinoma.

I only have two minor notes:

(1) in Chapter 2.3 Histological and immunohistochemical features...

"In immunohistochemical analysis, the antibodies CA19-9, EMA,S100, CD56, N-cadherin, and CRP showed potential in the differential diagnosis of iCCA subtypes, while CK7 and S100 did not (Table 3)."

S100 is listed in both arguments, whereas in Table 3, S100 Antibody is not shown as a helpful discriminator. Please clarify.

We thank this reviewer for this observation.  In our hands, immunohistochemical analysis of S100 was not a good discriminator between small and large duct iCCA.It is worth mentioning that we did not use the same S100 antibody as in the original publication, which may explain some discrepancy.

(2) in Chapter 3

"the diagnosis needs further l immunohistochemical analyses such as Hepatocyte paraffin 1 (HepPar1) [25]. "

I would expect the "l" being a typo

Congratulation to the nice piece of work,

Thank you very much for your kind words! We have corrected the typo.

Reviewer 3 Report

In the manuscript, the authors perfomred an integrative analysis of ICC subtypes for patients stratification in aspect of clinical, pathological and radiological results.  The paper is well written and the analysis is well conducted. There are several minor issues that should be addressed.

1. What is the diagnostic criteria of distinguishing small duct ICC and large duct ICC? Is it based on pathological or radiological findings? It should be presented in the methods part.

2. It is noted that the statistical methods used in the results were Cox proportional hazard model analyzing the univariate and multivariate factors.  Why not use Chi test comparing the correlations between subtype of ICC and categorical variables? 

Author Response

In the manuscript, the authors perfomred an integrative analysis of ICC subtypes for patients stratification in aspect of clinical, pathological and radiological results. The paper is well written and the analysis is well conducted. There are several minor issues that should be addressed.

  1. What is the diagnostic criteria of distinguishing small duct ICC and large duct ICC? Is it based on pathological or radiological findings? It should be presented in the methods part.

We thank the reviewer for his suggestion. The initial subtype diagnosis was decided after viewing all section specimens by two experienced pathologists with special expertise in the field of liver pathology (B.K.S. and T.S.G). (see paragraph 5.3). In order to avoid possible misunderstandings due to the structure of the material and methods part, we have added the following sentence under 5.1: “The subtype diagnosis was mainly based on a combination of macroscopic and histopathological para­meters.”

  1. It is noted that the statistical methods used in the results were Cox proportional hazard model analyzing the univariate and multivariate factors. Why not use Chi test comparing the correlations between subtype of ICC and categorical variables?

We fear that a misunderstanding has occurred in this respect, presumably due to the text structure. We performed a chi-squared test for the analysis of the categorical variables and determined Cramér's V accordingly to determine the effect size. We created a subsection to better separate the statistical analysis from the rest of the text.
